# Perioperative and anesthesia-related cardiac arrest and mortality rates in Brazil: A systematic review and proportion meta-analysis

Leandro G. Braz[1]*, José R. C. Braz[1], Marilia P. Modolo[1], Jose E. Corrente[2], Rafael Sanchez[1], Mariana Pacchioni[1], Julia B. Cury[1], Iva B. Soares[1], Mariana G. Braz[1]

**1** Department of Anesthesiology, Botucatu Medical School, Sao Paulo State University—UNESP, São Paulo, Brazil, **2** Department of Biostatistics, Institute of Biosciences, Sao Paulo State University—UNESP, São Paulo, Brazil

* leandro.braz@unesp.br

## Abstract

### Introduction

Studies have shown that both perioperative and anesthesia-related cardiac arrest (CA) and mortality rates are much higher in developing countries than in developed countries. This review aimed to compare the rates of perioperative and anesthesia-related CA and mortality during 2 time periods in Brazil.

### Methods

A systematic review with meta-analysis of full-text Brazilian observational studies was conducted by searching the Medline, EMBASE, LILACS and SciELO databases up to January 29, 2020. The primary outcomes were perioperative CA and mortality rates and the secondary outcomes included anesthesia-related CA and mortality events rates up to 48 postoperative hours.

### Results

Eleven studies including 719,273 anesthetic procedures, 962 perioperative CAs, 134 anesthesia-related CAs, 1,239 perioperative deaths and 29 anesthesia-related deaths were included. The event rates were evaluated in 2 time periods: pre-1990 and 1990–2020. Perioperative CA rates (per 10,000 anesthetics) decreased from 39.87 (95% confidence interval [CI]: 34.60–45.50) before 1990 to 17.61 (95% CI: 9.21–28.68) in 1990–2020 (P < 0.0001), while the perioperative mortality rate did not alter (from 19.25 [95% CI: 15.64–23.24] pre-1990 to 25.40 [95% CI: 13.01–41.86] in 1990–2020; P = 0.1984). Simultaneously, the anesthesia-related CA rate decreased from 14.39 (95% CI: 11.29–17.86) to 3.90 (95% CI: 2.93–5.01; P < 0.0001), while there was no significant difference in the anesthesia-related mortality rate (from 1.75 [95% CI: 0.76–3.11] to 0.67 [95% CI: 0.09–1.66; P = 0.5404).

**Data Availability Statement:** All relevant data are within the manuscript and its Supporting Information files.

**Funding:** R.S., M.P. and J.B.C. received a scholarship from The National Council for Scientific and Technological Development (CNPq; 125054/2016-5, 149852/2019-3 and 144230/2019-4, respectively) and L.G.B. received a fellowship from CNPq (304174/2018-1). M.P.M. received a scholarship from Coordination of Improvement for Higher Academic Staff (CAPES). Both CNPq and CAPES supported our study by financial support. We did not receive funding from our institution. The funders had no role in study design, data collection and analysis, decision to publish, or preparation of the manuscript.

**Competing interests:** The authors have declared that no competing interests exist.

## Conclusions

This review demonstrates an important reduction in the perioperative CA rate over time in Brazil, with a large and consistent decrease in the anesthesia-related CA rate; however, there were no significant differences in perioperative and anesthesia-related mortality rates between the assessed time periods.

## Introduction

The global surgical volume is increasing worldwide; however, the safety and quality of surgical care and anesthesia management remain poor in many regions, and outcome assessments are not often prioritized in developing countries [1]. Among the complications of surgery, cardiac arrest (CA) is one of the worst events, as it can result in sequelae, loss of function and death. The rates of perioperative CA and mortality can be used to explore the differences among facilities that perform surgical and anesthesia procedures in different centers and can serve as quality indicators to promote improvements in patient safety and reductions in unfavorable outcomes [2–4]. Brazil is a developing country that, over the years, has experienced great difficulty in offering adequate healthcare to its population [5].

Two systematic reviews with a proportion meta-analysis of worldwide studies have shown that the rates of both perioperative and anesthesia-related CA and mortality are much higher in developing countries than in developed countries [6, 7]. A review of the studies conducted in Brazil showed that there was a reduction in the rate of perioperative CA during the last 25 years, similar to the global trend [8].

This systematic review aimed to compare the rates of perioperative and anesthesia-related CA and mortality during 2 time periods in Brazil. We hypothesized that the rates of intraoperative and anesthesia-related CA and mortality have decreased over time.

## Methods

The Cochrane Handbook for Intervention Reviews [9] guided our choice of methods. Our report adheres to the Preferred Reporting Items for Systematic Reviews and Meta-Analyses (PRISMA) statement (S1 File) [10]. The protocol for this proportional meta-analysis was registered in a public registry (PROSPERO, #CRD42019141158).

### Search strategy

We searched the medical literature to identify all Brazilian observational studies that reported perioperative and/or anesthesia-related CA or mortality rates. We searched the Medline (PubMed), EMBASE, Latin American and Caribbean Health Sciences Literature (LILACS), and Scientific Electronics Library Online (SciELO) databases from their inception to January 29, 2020. Search strategies were developed in consultation with a research librarian with systematic review expertise. The search was conducted using Index Terms (e.g., MeSH and Emtree) and text words and word variants for "an(a)esthesia", "cardiac arrest" and "mortality", as well as an exhaustive list of synonyms. The search strategy was adapted to each database to find relevant studies (S2 File). Five independent reviewers (L.G.B., M.P.M., R.S., M.P., and J.B.C.) performed the initial screening of the titles and abstracts of the studies to identify whether they contained relevant content. The relevant studies were downloaded and reviewed in full by 2 independent reviewers (L.G.B. and I.B.S.). These reviewers also reviewed the references of

the included studies and added additional relevant studies. There were no restrictions on either year of publication or language.

## Outcome definitions

The primary outcomes were perioperative CA and mortality events (regardless of triggering factors: patient's disease/condition, surgery and anesthesia). The secondary outcomes were anesthesia-related CA and mortality events (entirely and partially anesthesia-related), and entirely anesthesia-related CA and mortality events, defined as CA or mortality deemed to be attributable only to anesthesia (e.g., depression of ventilation leading to hypoxemic CA after anesthesia induction in a stable patient without comorbidities). All perioperative, anesthesia-related, and entirely anesthesia-related CAs or mortality were defined by the authors of the studies included in this review.

## Selection criteria

To ensure that the studies represented a mixed population, full-text studies were included if they fulfilled the following criteria: (1) observational studies; and (2) the study's outcome measurements were perioperative and/or anesthesia-related CA or perioperative and/or anesthesia-related mortality up to 48 postoperative hours.

Studies were excluded if they met any of the following criteria: (1) the study focused on specific age groups (e.g., only geriatric patients); (2) the study included only a specific anesthetic technique (e.g., only neuraxial blockade); (3) the study included only a specific surgery type (e.g., only cardiac surgery); (4) the study included only a specific American Society of Anesthesiologists (ASA) physical status (e.g., only ASA I patients); and (5) the study included less than 3,000 patients because this number is the minimum required to enable estimation of the incidence of rare adverse events ($\leq$ 1 per 1,000 anesthetics), according to the rule of three sample size approximations [11].

## Assessment of methodological quality

Two independent reviewers (L.G.B. and J.R.C.B.) assessed the studies selected for retrieval for methodological validity prior to inclusion in the review using the Joanna Briggs Institute (JBI) Critical Appraisal Tool for Prevalence Studies [12]. The nine domains in this tool were target population, sampling, sample size, description of participants and setting, coverage of identified sample, methods of identifying the outcome, reliability of the outcome measurement, appropriate statistical analysis and response rate. The cutoff for inclusion/exclusion was determined through consensus between the reviewers. The cutoff point for the inclusion of a study in the review was a "yes" answer to at least five of the nine questions (more than 50%) on the standardized critical appraisal instrument. Discrepancies were resolved after discussion between the two authors; if this was not possible, the discrepancies were resolved with a third author (M.G.B.).

## Data extraction

We used a standard form to extract the data from the included studies (S1 Table). Two of the authors (L.G.B. and I.B.S.) independently identified the studies that were included in the review. Any discrepancies were resolved after discussion with a third reviewer (J.R.C.B.). When more than one study was published with the same population, data were extracted from the most recent and/or complete study. If eligible articles were missing data, their authors were contacted for clarification.

## Statistical analysis

**Proportion meta-analysis.** Using StatsDirect software (StatsDirect Ltd., Altrincham, Cheshire, UK), a random-effects model with inverse Freeman-Tukey double arcsine transformation was applied to calculate the weighted event rates across all of the studies included in the proportion meta-analysis [13, 14]. For the purpose of this study, the data were dichotomized into 2 time periods. A similar method was used in three prior systematic reviews: two on perioperative CA [6, 15] and one on perioperative CA and mortality [7]. This stratification was based on many safety-improvement measures, such as monitoring for oxygenation and ventilation parameters, anesthesia workstations with modern ventilators, new anesthesia medications, supraglottic devices, anesthesia and surgery safety protocols, advances in postoperative pain management, and an increase in the number of postanesthesia and intensive care beds, which emerged towards the end of the 1980s in developed countries and later in some developing countries [16, 17].

We performed a sensitivity analysis to define the cutoff year between the 2 time periods (from 1988 to 1992). The studies were allocated to one of the two periods based on their median patient recruitment interval [6, 7, 15]. The event rate was defined as the number of CAs or deaths per 10,000 anesthetics. The data are reported with their 95% confidence intervals (CIs). To compare differences in the proportions of events, a model with a binomial distribution adjusted for overdispersion was generated using SAS for Windows® software, v.9.4 (SAS Institute, Cary, NC). The $I^2$ statistic was also used as an alternative approach to quantify the degree of heterogeneity among studies [18]; values higher than 40% suggested significant heterogeneity among the studies [19]. A P value < 0.05 was considered significant.

## Results

### Selection of studies

The literature search identified 15,265 potentially eligible articles. After reviewing the titles and abstracts, we excluded 5,670 duplicate studies using the software EndNote® (X9.2/2019 version, Clarivate Analytics, PA) and 9,580 studies because of a lack of relevance. We retrieved 15 potentially relevant full-text papers for detailed evaluations. Of these articles, 11 studies met the inclusion criteria, including 6 studies reporting perioperative CA, 4 studies reporting anesthesia-related CA, 9 studies reporting perioperative mortality, and 7 studies reporting anesthesia-related mortality (Fig 1).

### Study characteristics

The earliest study included was published in 1986 [20], and the most recent was published in 2019 [21]. The studies included 719,273 anesthetic procedures with 962 perioperative CA events, 134 anesthesia-related CA events, 1,239 perioperative deaths and 29 anesthesia-related deaths. The characteristics and designs of the studies are presented in Tables 1 and 2. The forest plots summarizing the data are presented in S3 File. As expected, there was significant heterogeneity among the studies, with a *minimum $I^2$* of 6.5% and a *maximum* of 98.3% for CA and a *minimum $I^2$* of 49.1% and a *maximum* of 99.0% for mortality (Tables 3 and 4).

### Methodological quality

Eleven studies meeting the inclusion criteria were assessed for methodological quality. All of them obtained more than five "yes" answers, indicating that they were of good quality (S2 Table).

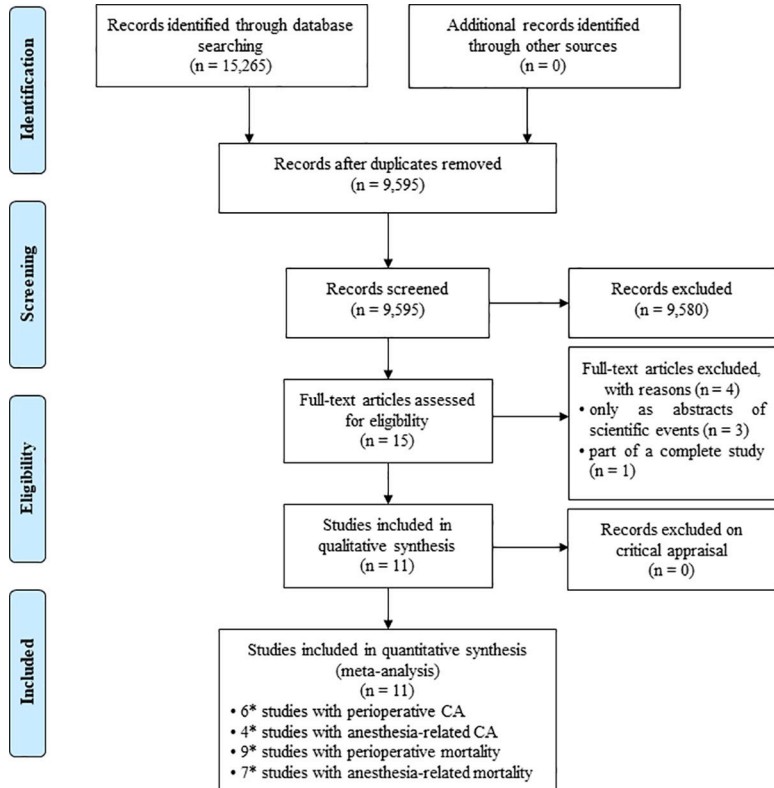

**Fig 1. PRISMA flow diagram showing the study selection process.** *Abbreviations*. CA: cardiac arrest; *some studies have been included in more than one category.

## Proportion meta-analysis of CA rate

The sensitivity analysis revealed no substantial change in the event rates during the time period from 1988 to 1992, justifying the establishment of 1990 as the temporal cutoff point. Thus, the event rates during 2 time periods (pre-1990 *versus* 1990–2020) were analyzed.

The rate of perioperative CA decreased (2.26-fold) from 39.87 (95% CI: 34.60 to 45.50) before 1990 to 17.61 (95% CI: 9.21 to 28.68) per 10,000 anesthetics in 1990–2020 (P < 0.0001) (Table 3). At the same time, the rates of both anesthesia-related and entirely anesthesia-related CA decreased consistently (by 3.7- and 12-fold, respectively) from 14.39 (95% CI: 11.29 to 17.86) and 7.00 (95% CI: 4.89 to 9.49) before 1990 to 3.90 (95% CI: 2.93 to 5.01) and 0.58 (95% CI: 0.00 to 3.72) per 10,000 anesthetics in 1990–2020, respectively (P < 0.0001 and P < 0.0001, respectively) (Table 3).

## Proportion meta-analysis of mortality rate

There was no significant alteration in the rate of either perioperative mortality (from 19.25 [95% CI: 15.64 to 23.24] before 1990 to 25.40 [95% CI: 13.01 to 41.86] per 10,000 anesthetics in 1990–2020; P = 0.1984) or anesthesia-related mortality (from 1.75 [95% CI: 0.76 to 3.11] before 1990 to 0.67 [95% CI: 0.09 to 1.66] per 10,000 anesthetics in 1990–2020; P = 0.5404; Table 4). None of the Brazilian studies evaluated entirely anesthesia-related mortality rates before the 1990s, making a subanalysis of data between periods impossible; this rate in 1990–2020 was 0.22 (95% CI: 0.05 to 0.68) per 10,000 anesthetics (Table 4).

**Table 1. Description of included cardiac arrest studies.**

| Investigators and year of publication | Data source, study period, and median year | Time period of CA occurrence | Excluded | Patients (n) | Perioperative CA (n) | Anesthesia-related CA (n) | Entirely anesthesia-related CA (n) |
|---|---|---|---|---|---|---|---|
| Ruiz Neto & Amaral, 1986 [20] | Tertiary university hospital Medical records 1982–1984 | OR | Cardiac surgery | 51,422 | 205 | 74 | 36 |
| | 1983 | | | | | | |
| Braz et al., 1999 [22] | Tertiary university hospital —Database | OR and PACU | - | 58,553 | 184 | 21 | - |
| | 1988–1996 | | | | | | |
| | 1992 | | | | | | |
| Braz et al., 2006 [23] | Tertiary university hospital —Database | OR and PACU | - | 53,718 | - | 18 | 10 |
| | 1996–2005 | | | | | | |
| | 2001 | | | | | | |
| Sebbag et al., 2013 [24] | Tertiary university hospital —Database 2007 | OR | Cardiac surgery | 40,379 | 52 | 21 | 0 |
| | 2007 | | | | | | |
| Toledo et al., 2013 [25] | Tertiary university hospital —Database 2007–2009 | OR | Cardiac surgery | 81,587 | 81 | - | - |
| | | | < 18 years | | | | |
| | 2008 | | | | | | |
| Carlucci et al., 2014 [26] | Tertiary university hospital —Database | OR and PACU | - | 90,909 | 280 | - | - |
| | 1996–2009 | | | | | | |
| | 2003 | | | | | | |
| Vane et al., 2019 [21] | Tertiary university hospital —Database 2007–2014 | OR | Cardiac surgery | 167,574 | 160 | - | - |
| | | | < 18 years | | | | |
| | 2011 | | | | | | |

*Abbreviations.* CA: cardiac arrest; OR: operating room; PACU: postanesthesia care unit

## Discussion

To the best of our knowledge, this study is the first comprehensive synthesis showing the temporal trends in perioperative CA and mortality rates in a developing country. When comparing the pre-1990 and 1990–2020 periods in Brazilian studies, a reduction in the perioperative CA rates was found, with a relatively large decrease in the rate of anesthesia-related CA; however, there were no significant differences in the rates of perioperative and anesthesia-related mortality.

A systematic review with a proportion meta-analysis of the worldwide studies from developed and developing countries during 2 time periods (pre-1990 *versus* 1990–2014) showed a significant increase in the rate of perioperative CA and no differences in the rates of anesthesia-related and entirely anesthesia-related CA in developing countries, while in developed countries, there was a significant decrease in all CA rates [6], similar to the findings of the present study. Comparing the CA rates after 1990, we verified that the rates of both perioperative and anesthesia-related CA in Brazilian studies were similar to the rates in developing countries (19.9 and 4.5 per 10,000 anesthetics, respectively) but were 2.8- and 5.6-fold higher, respectively, than the rates in developed countries (6.2 and 0.7 per 10,000 anesthetics, respectively); the rate of entirely anesthesia-related CA in Brazilian studies was 2.7-fold lower than the rate

**Table 2. Description of included mortality studies.**

| Investigators | Data source, study period, and median year | Time period in which death occurred | Excluded | Patients (n) | Perioperative mortality (n) | Anesthesia-related mortality (n) | Entirely anesthesia-related mortality (n) |
|---|---|---|---|---|---|---|---|
| Ruiz Neto & Amaral, 1986 [20] | Tertiary university hospital / Medical records / 1982–1984 / 1983 | OR | Cardiac surgery | 51,422 | 99 | 9 | - |
| Cicarelli et al., 1998 [27] | Tertiary university hospital—Database / 1995 / 1995 | Up to 24 hours postoperative | Cardiac surgery | 25,926 | 129 | 2 | 2 |
| Braz et al., 1999 [22] | Tertiary university hospital—Database / 1988–1996 / 1992 | OR and PACU | - | 58,553 | 124 | 5 | - |
| Chan & Auler Jr, 2002 [28] | Tertiary university hospital—Database / 1998–1999 / 1999 | Up to 24 hours postoperative | - | 82,641 | 424 | 1 | 1 |
| Braz et al., 2006 [23] | Tertiary university hospital—Database / 1996–2005 / 2001 | OR and PACU | - | 53,718 | - | 6 | 3 |
| Sebbag et al., 2013 [24] | Tertiary university hospital—Database / 2007 / 2007 | Up to 24 hours postoperative | Cardiac surgery | 40,379 | 32 | - | - |
| Carlucci et al., 2014 [26] | Tertiary university hospital—Database / 1996–2009 / 2003 | OR and PACU | - | 90,909 | 181 | - | - |
| Pignaton et al., 2016 [29] | Tertiary university hospital—Database / 2005–2012 / 2009 | OR and PACU | - | 55,002 | 88 | 0 | 0 |
| Stefani et al., 2018 [30] | Quaternary university hospital -Database / 2012–2013 / 2013 | Up to 48 hours postoperative | - | 11,562 | 76 | 6 | 1 |
| Vane et al., 2019 [21] | Tertiary university hospital—Database / 2007–2014 / 2011 | Up to 24 hours postoperative | Cardiac surgery / < 18 years | 167,574 | 86 | - | - |

*Abbreviations.* OR: operating room; PACU: postanesthesia care unit

in developing countries (1.6 per 10,000 anesthetics) and similar to the rate in developed countries (0.5 per 10,000 anesthetics) in the aforementioned review.

Contrary to the findings of Brazilian studies, a systematic review with a meta-analysis of studies from developed and developing countries showed a significant reduction in the rate of

**Table 3. Proportion meta-analysis of cardiac arrest rates in Brazilian studies by time period.**

| Time period | Studies n | $I^2$% | Events n | Patients n | Proportion meta-analysis per 10,000 anesthetics (95% CI) | P value for subgroup Pre-1990 *versus* 1990–2020 |
|---|---|---|---|---|---|---|
| Perioperative cardiac arrest | | | | | | |
| **Pre-1990** | 1 | NA | 205 | 51,422 | 39.87 (34.60–45.50) | < 0.0001 |
| **1990–2020** | 5 | 98.3 | 757 | 439,002 | 17.61 (9.21–28.68) | |
| Anesthesia-related cardiac arrest | | | | | | |
| **Pre-1990** | 1 | NA | 74 | 51,422 | 14.39 (11.29–17.86) | < 0.0001 |
| **1990–2020** | 3 | 6.5 | 60 | 152,650 | 3.90 (2.93–5.01) | |
| Entirely anesthesia-related cardiac arrest | | | | | | |
| **Pre-1990** | 1 | NA | 36 | 51,422 | 7.00 (4.89–9.49) | < 0.0001 |
| **1990–2020** | 2 | 91.8 | 10 | 94,097 | 0.58 (0.00–3.72) | |

*Abbreviations.* $I^2$: indicates heterogeneity among studies; CI = confidence interval; NA = not available

perioperative mortality over 3 time periods (pre-1970s, 1970s-1980s and 1990–2009), despite the increasing baseline ASA risk status of the patients [7]. The authors of this review suggested that their results indicated improved perioperative and anesthesia safety, particularly in developed countries. Comparing the mortality rates between the aforementioned review and Brazilian studies published after 1990, we verified that the rate of perioperative mortality in Brazil was similar to the rate in developing countries (24.45 per 10,000 anesthetics) and 2.3-fold higher than the rate in developed countries (10.95 per 10,000 anesthetics), while the rate of entirely anesthesia-related mortality in Brazil was 6.4-fold lower than the rate in developing countries (1.41 per 10,000 anesthetics) and similar to the rate in developed countries (0.25 per 10,000 anesthetics). The rate of anesthesia-related mortality was 2.4-fold higher than the rate (0.28 per 10,000 anesthetics) reported in a recent study in the USA [31].

There is no consensus about the time frame for perioperative CA or mortality events [32]. Thus, perioperative CA and mortality rates depend on how the perioperative period is defined: only intraoperative, intraoperative and recovery from anesthesia, up to 24 h postoperative or up to 48 h postoperative. It must be highlighted that all CA events reported in the studies occurred in only the intraoperative period (operating room or operating room plus postanesthesia care unit), while in the mortality studies, the events occurred in the intraoperative

**Table 4. Proportion meta-analysis of mortality rates in Brazilian studies by time period.**

| Time period | Studies n | $I^2$% | Events n | Patients n | Proportion meta-analysis per 10,000 anesthetics (95% CI) | P value for subgroup Pre-1990 *versus* 1990–2020 |
|---|---|---|---|---|---|---|
| Perioperative mortality | | | | | | |
| **Pre-1990** | 1 | NA | 99 | 51,422 | 19.25 (15.64–23.24) | 0.1984 |
| **1990–2020** | 8 | 99.0 | 1,054 | 532,546 | 25.40 (13.01–41.86) | |
| Anesthesia-related mortality | | | | | | |
| **Pre-1990** | 1 | NA | 9 | 51,422 | 1.75 (0.76–3.11) | 0.5404 |
| **1990–2020** | 6 | 81.0 | 20 | 287,402 | 0.67 (0.09–1.66) | |
| Entirely anesthesia-related mortality | | | | | | |
| **Pre-1990** | - | - | - | - | - | NA |
| **1990–2020** | 5 | 49.1 | 7 | 228,849 | 0.22 (0.05–0.68) | |

*Abbreviations.* $I^2$: indicates heterogeneity among studies; CI = confidence interval; NA = not available

(55.6%) or 24–48 h postoperative (44.4%) periods. A Brazilian study reported a postoperative 24 h mortality rate (7.9 per 10,000 anesthetics) that was 2-fold higher than the intraoperative mortality rate (3.9 per 10,000 anesthetics) [24], while a Korean study reported a postoperative 24 h mortality rate (1.25 per 10,000 anesthetics) that was 8-fold higher than the intraoperative mortality rate (0.15 per 10,000 anesthetics) [33]. Thus, the time frame for the events may have interfered with the results of the current review.

Poor ASA physical status (III-V) has been reported to be the most important predictor of perioperative CA and mortality events in Brazilian studies [23, 24, 28, 29] and in studies from developed countries [31, 34, 35]. Two Brazilian studies demonstrated that many patients present for surgery with poor health at baseline and without the optimization of disease management [29, 36]. Thus, patient disease/condition remained the major triggering factor for perioperative CA and mortality, followed by surgery and anesthesia in low proportions [23, 28, 29].

The most influential cause of perioperative CA and mortality events in Brazilian studies was sepsis, followed by trauma [23, 29, 30, 37]. Therefore, comorbid conditions seem to be major contributors to the high perioperative CA and mortality rates in Brazil [5]. Thus, preanesthetic management of comorbidities plays a major role in minimizing perioperative complications and adverse effects. These findings demonstrate that there is a need to improve the quality and quantity of resources that can be used as well as access to healthcare, both of which are inadequate, in developing countries [15].

In Brazil and other developing countries, the combination of poverty and precarious healthcare with increasing and aging populations has increased both the number of patients in poor physical condition and the demand for surgical procedures in recent decades. However, the surgical volume in Brazil is approximately 2.5-fold lower [38, 39] than the international target set to achieve universal access to surgical care, which is 5 surgeries per 100 inhabitants per year [4]. In addition, in 2014, the density of the health professionals (surgeons, anesthesiologists and obstetricians) was 34.7 per 100,000 inhabitants in Brazil [38], while developed countries have an average of 56.9 of these professionals per 100,000 inhabitants [40]. These factors, combined with limited resources and numbers of surgical beds and operating rooms and increasing costs of surgery and anesthesia, seem to result in an important consequence: high perioperative CA and mortality rates in developing countries [41].

Governmental and nongovernmental organizations should prioritize and increase healthcare investments in Brazil and other developing countries [42]. Policy makers and healthcare professionals must address practices that have demonstrable effectiveness in improving perioperative outcomes. Human resources are pivotal; the number and the education and training of both anesthesiologists and surgeons must be increased [41, 42]. Preoperative management must be optimized; multidisciplinary discussions of adverse effects must be prompted; the provision of new monitoring techniques (e.g., pulse oximetry, capnography, echocardiography), modern anesthetic drugs, anesthesia equipment and workstations must be addressed universally; and practice guidelines and checklists in surgery and anesthesia as well as a structured approach to reducing errors must be adopted [41–45]. The mandatory period in the postanesthesia care unit should also be expanded, and the number of intensive care beds for critical patients should be increased to minimize the occurrence of adverse events. Global efforts should be directed towards ensuring that the increase in the surgical volume is accompanied by the implementation of basic safety measures in developing countries. Feedback regarding the implementation of such measures should be required from both developed and developing countries to ensure that the existing gap between health care systems is effectively reduced [1].

The limitations of this review must be acknowledged. This review is limited by a paucity of studies, particularly before 1990, which could have influenced the findings. Thus, we were not

able to analyze publication bias because there were fewer than ten eligible studies addressing each outcome in a proportion meta-analysis; otherwise, the findings from the assessment of risk of bias would not be reliable. Despite the heterogeneity and small number of studies, the meta-analyses showed consistent results that strongly reinforce the relevance of our systematic review. The studies differed greatly in their design; differences in the time frames of the events (e.g., intraoperative or up to 24–48 h postoperative) and in the types of surgery (e.g., whether cardiac surgeries were included) accounted for most of the heterogeneity. A random-effects model with inverse Freeman-Tukey double arcsine transformation was applied to minimize this heterogeneity when assessing the trends between the two time periods. All the studies were conducted at only governmental tertiary or quaternary university hospitals. Some studies included a small sample size, and all included studies presented data from a single center; there were no multicenter studies. We included only studies with > 3,000 patients and calculated weighted event rates across all studies to minimize possible bias. The case mix, the type of anesthesia and surgery, and ASA physical status classification were not included in this review. Considering that the current review covered a 32-year period with recruitment of patients from 1982 to 2014, it must be highlighted that significant advances in resuscitation, such as, Advanced Trauma Life Support (ATLS) and Advanced Cardiovascular Life Support (ACLS), and operative protocols have occurred, which may have influenced the results of the current review.

This review demonstrates that the rates of perioperative CA have decreased over time in Brazil, and the proportional decrease has been large and consistent in the rates of anesthesia-related and entirely anesthesia-related CA. However, the rates of both perioperative and anesthesia-related mortality did not show significant differences between the time periods. Further reviews of perioperative and anesthesia-related CA and mortality must be periodically performed to continue to monitor the rates in surgical patients in Brazil.

## Supporting information

**S1 File. PRISMA checklist.**
(DOC)

**S2 File. Search strategy.**
(DOCX)

**S3 File. Forest plots.**
(DOCX)

**S1 Table. Standard form used to extract information from the studies.**
(DOCX)

**S2 Table. Critical appraisal results for included studies.**
(DOCX)

## Author Contributions

**Conceptualization:** Leandro G. Braz, José R. C. Braz.

**Data curation:** Leandro G. Braz.

**Formal analysis:** Leandro G. Braz, Marilia P. Modolo, Rafael Sanchez, Mariana Pacchioni, Julia B. Cury.

**Investigation:** Leandro G. Braz, José R. C. Braz, Marilia P. Modolo, Rafael Sanchez, Mariana Pacchioni, Julia B. Cury, Iva B. Soares.

**Methodology:** Leandro G. Braz, José R. C. Braz, Marilia P. Modolo, Jose E. Corrente, Rafael Sanchez, Mariana Pacchioni, Julia B. Cury, Iva B. Soares.

**Project administration:** Leandro G. Braz, José R. C. Braz.

**Supervision:** Leandro G. Braz, José R. C. Braz.

**Validation:** Leandro G. Braz, Jose E. Corrente.

**Visualization:** Leandro G. Braz, Mariana G. Braz.

**Writing – original draft:** Leandro G. Braz, José R. C. Braz, Mariana G. Braz.

**Writing – review & editing:** Leandro G. Braz, José R. C. Braz, Marilia P. Modolo, Jose E. Corrente, Rafael Sanchez, Mariana Pacchioni, Julia B. Cury, Iva B. Soares, Mariana G. Braz.

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
