## [Decision Letter · Decision Letter 0]

19 Jun 2020

PONE-D-20-08860

Perioperative and anesthesia-related cardiac arrest and mortality rates in Brazil: a systematic review and proportion meta-analysis

PLOS ONE

Dear Dr. Braz,

Thank you for submitting your manuscript to PLOS ONE. After careful consideration, we feel that it has merit but does not fully meet PLOS ONE’s publication criteria as it currently stands. Therefore, we invite you to submit a revised version of the manuscript that addresses the points raised during the review process.

We look forward to receiving your revised manuscript.

Kind regards,

Omid Beiki, M.D., Ph.D.

Academic Editor

PLOS ONE

Journal Requirements:

2. In the Methods, please specify any assessment of risk of bias that may affect the cumulative evidence (e.g., publication bias, selective reporting within studies). Please ensure that the specific method of assessment (funnel plot, Egger's test, Begg's test, etc) is mentioned.

3. In the Methods, please specify how study quality was assessed.

R.S., M.P. and J.B.C. received a scholarship from The National Council for

Scientific and Technological Development (CNPq) and L.G.B. received a fellowship

from CNPq (304466/2015-8). M.P.M. received a scholarship from Coordination of

Improvement for Higher Academic Staff (CAPES). CNPq and CAPES are Brazilian

governmental agency dedicated for promoting scientific research.

The authors received no specific funding for this work.

Reviewers' comments:

Reviewer's Responses to Questions

**Comments to the Author**

1. Is the manuscript technically sound, and do the data support the conclusions?

Reviewer #1: Partly

Reviewer #2: Partly

2. Has the statistical analysis been performed appropriately and rigorously? 

Reviewer #1: I Don't Know

Reviewer #2: Yes

3. Have the authors made all data underlying the findings in their manuscript fully available?

Reviewer #1: Yes

Reviewer #2: Yes

4. Is the manuscript presented in an intelligible fashion and written in standard English?

Reviewer #1: No

Reviewer #2: Yes

5. Review Comments to the Author

Reviewer #1: Thank you for the opportunity to review this manuscript.

I have a few comments and questions for the authors.

Under methods; you've indicated you adhered to the MOOSE guidelines, but this is not a methodology, this is a guideline that serves as a checklist for reporting observational studies in systematic reviews and meta-analyses. What methodological framework, or handbook did you use to conduct this systematic review and meta-analysis? The Cochrane Handbook? JBI? CRD York? You must always state the methods you followed to conduct this study, in addition to the relevant reporting guideline, which you must use to report your study, in this case it would be PRISMA.

My recommendation would be that authors indicate the methods they used to conduct the study, and include the PRISMA checklist indicating where each of the items in the guideline was reported.

Some of the diagrams are confusing, primarily due to labelling, e.g. the references workflow diagram is your PRISMA Diagram, but it's not labelled adequately.

The manuscript could also benefit from copyediting, there are instances where the syntax is unclear.

Reviewer #2: Review of 'Perioperative and anesthesia-related cardiac arrest and mortality rates...'

This paper is a review and meta-analysis whose goal is to summarize the evidence surrounding changes in perioperative and anesthesia-related cardiac arrest and mortality rates in Brazil.

I am not an expert on cardiac arrest and/or surgery in Brazil, so I can only comment on the statistical part of this paper.

The writing is clear and I think it is possible from this text to evaluate what the statistical analysis that the authors have performed.

My main concern is that the meta-analysis reveals quite large amounts of heterogeneity for several important quantities; Tables 3 and 4 suggest that I2 can be as high as 99 percent, and is higher than 80 percent for 4 out of 6 of the quantities. This is somewhat subjective, but I am not confident that a meta-analysis can be meaningfully performed if there is this much heterogeneity in the effects being aggregated. The authors might at least consider reporting additional statistics intended to help the reader understand how much heterogeneity there is in the estimated true effect sizes.

Minor comments:

* pg 5 - I believe this should read 'sample size' instead of 'simple size'

6. PLOS authors have the option to publish the peer review history of their article (what does this mean?). If published, this will include your full peer review and any attached files.

Reviewer #1: No

Reviewer #2: No

---

## [Author Response · Author response to Decision Letter 0]

24 Jul 2020

Answers to the Editor

Answer: First, we would like to thank the Editor for taking the time to review our manuscript and make important comments and suggestions. We tried our best to answer the questions and make the required changes in the manuscript after considering the suggestions, which certainly improved the revised manuscript.

1. “Please ensure that your manuscript meets PLOS ONE's style requirements, including those for file naming. The PLOS ONE style templates can be found at

https://journals.plos.org/plosone/s/file?id=wjVg/PLOSOne_formatting_sample_main_body.pdf and https://journals.plos.org/plosone/s/file?id=ba62/PLOSOne_formatting_sample_title_authors_affiliations.pdf”

Answer: We apologize. Our revised manuscript meets PLOS ONE´s style requirements.

2. “In the Methods, please specify any assessment of risk of bias that may affect the cumulative evidence (e.g., publication bias, selective reporting within studies). Please ensure that the specific method of assessment (funnel plot, Egger's test, Begg's test, etc) is mentioned.”

Answer: We agree with the suggestion, but we would like to clarify that our review is limited by a paucity of studies. Thus, unfortunately, we were not able to analyze publication bias because there were fewer than ten eligible studies addressing each outcome in the meta-analysis; otherwise, the findings from the assessment of risk of bias would not be reliable. Despite the heterogeneity and small number of studies, the meta-analyses showed consistent results that strongly reinforce the relevance of our systematic review. Thus, we included the information as one of the limitations of our review in the discussion section.

3. “In the Methods, please specify how study quality was assessed.”

Answer: We agree with the comment. We have added the topic “Assessment of methodological quality” to the methods, specifying the assessment of study quality using the Joanna Briggs Institute (JBI) - Critical Appraisal Tool for Prevalence Studies.

4. “Thank you for stating the following in the Acknowledgments Section of your manuscript:

R.S., M.P. and J.B.C. received a scholarship from The National Council for Scientific and Technological Development (CNPq) and L.G.B. received a fellowship from CNPq (304466/2015-8). M.P.M. received a scholarship from Coordination of Improvement for Higher Academic Staff (CAPES). CNPq and CAPES are Brazilian governmental agency dedicated for promoting scientific research.

We note that you have provided funding information that is not currently declared in your Funding Statement. However, funding information should not appear in the Acknowledgments section or other areas of your manuscript. We will only publish funding information present in the Funding Statement section of the online submission form. Please remove any funding-related text from the manuscript and let us know how you would like to update your Funding Statement. Currently, your Funding Statement reads as follows: The authors received no specific funding for this work.”

Answer: We apologize, and we have corrected the funding statement.

5. “Please include captions for your Supporting Information files at the end of your manuscript, and update any in-text citations to match accordingly. Please see our Supporting Information guidelines for more information: http://journals.plos.org/plosone/s/supporting-information.”

Answer: The captions for the supporting information files are at the end of our revised manuscript.

Answers to the Reviewers

Reviewer #1

“Thank you for the opportunity to review this manuscript.”

Answer: We would like to thank you for taking the time to review our manuscript and to make important suggestions. We tried our best to answer the questions and make the required changes in the manuscript after considering the suggestions, which certainly improved the revised manuscript.

1. “I have a few comments and questions for the authors. Under methods; you've indicated you adhered to the MOOSE guidelines, but this is not a methodology, this is a guideline that serves as a checklist for reporting observational studies in systematic reviews and meta-analyses. What methodological framework, or handbook did you use to conduct this systematic review and meta-analysis? The Cochrane Handbook? JBI? CRD York? You must always state the methods you followed to conduct this study, in addition to the relevant reporting guideline, which you must use to report your study, in this case it would be PRISMA. My recommendation would be that authors indicate the methods they used to conduct the study, and include the PRISMA checklist indicating where each of the items in the guideline was reported. Some of the diagrams are confusing, primarily due to labelling, e.g. the references workflow diagram is your PRISMA Diagram, but it's not labelled adequatey.”

Answer: As suggested by the reviewer, we described in the first paragraph of the methods section that The Cochrane Handbook for Intervention Reviews guided our choice of methods. Our reporting adheres to the Preferred Reporting Items for Systematic Reviews and Meta-analyses (PRISMA) statement. We followed the PRISMA diagram, as requested. 

2. “The manuscript could also benefit from copyediting, there are instances where the syntax is unclear.”

Answer: We apologize. The original version of the manuscript was copyedited by a native English speaker. The revised manuscript was copyedited again and improved.

Reviewer #2

1. “This paper is a review and meta-analysis whose goal is to summarize the evidence surrounding changes in perioperative and anesthesia-related cardiac arrest and mortality rates in Brazil.” 

Answer: We would like to thank you for taking the time to review our manuscript and to make important comments. We tried our best to answer the comments and to clarify the revised manuscript.

2) “I am not an expert on cardiac arrest and/or surgery in Brazil, so I can only comment on the statistical part of this paper. The writing is clear and I think it is possible from this text to evaluate what the statistical analysis that the authors have performed.

My main concern is that the meta-analysis reveals quite large amounts of heterogeneity for several important quantities; Tables 3 and 4 suggest that I2 can be as high as 99 percent, and is higher than 80 percent for 4 out of 6 of the quantities. This is somewhat subjective, but I am not confident that a meta-analysis can be meaningfully performed if there is this much heterogeneity in the effects being aggregated. The authors might at least consider reporting additional statistics intended to help the reader understand how much heterogeneity there is in the estimated true effect sizes.”

Answer: We agree with the comment that most of the proportion meta-analyses are heterogeneous. As previously described in the original manuscript (limitation paragraph of the discussion section), a random-effects model with inverse Freeman-Tukey double arcsine transformation (Freeman-Tukey transformation) was applied to minimize this heterogeneity when assessing the trends between the two time periods. If we had used a fixed-effects model (please see the table below), the results of the event rates and the statistical analyses (P values) would be different and incorrect. Thus, we were aware of the high degree of heterogeneity, and we applied the best options to minimize it following Higgins JP al.’s (2003) instructions - “Measuring inconsistency in meta-analyses”.

Table. Proportion meta-analysis of mortality rates in Brazilian studies by time period

Time period Studies

n I2 

% Events

n Patients

n Proportion meta-analysis 

per 10,000 anesthetics

 (95% CI) P value for subgroup

 Pre-1990 versus 1990-2020

Anesthesia-related mortality - Random-effects model (included in the manuscript) 

Pre-1990 1 NA 9 51,422 1.75 (0.76-3.11) 0.5404

1990-2020 6 81.0 20 287,402 0.67 (0.09-1.66) 

Anesthesia-related mortality - Fixed-effects model (incorrect presentation) 

Pre-1990 1 NA 9 51,422 1.75 (0.76-3.11) < 0.05

1990-2020 6 81.0 20 287,402 0.41 (0.18-0.71) 

3) “Minor comments: * pg 5 - I believe this should read 'sample size' instead of 'simple size'”

Answer: We apologize for our mistake. We corrected the word to “sample”.

---

## [Decision Letter · Decision Letter 1]

16 Oct 2020

PONE-D-20-08860R1

Perioperative and anesthesia-related cardiac arrest and mortality rates in Brazil: a systematic review and proportion meta-analysis

PLOS ONE

Dear Dr. Braz,

Thank you for submitting your manuscript to PLOS ONE. After careful consideration, we feel that it has merit but does not fully meet PLOS ONE’s publication criteria as it currently stands. Therefore, we invite you to submit a revised version of the manuscript that addresses the points raised during the review process.

We look forward to receiving your revised manuscript.

Kind regards,

Omid Beiki, M.D., Ph.D.

Academic Editor

PLOS ONE

Reviewers' comments:

Reviewer's Responses to Questions

**Comments to the Author**

1. If the authors have adequately addressed your comments raised in a previous round of review and you feel that this manuscript is now acceptable for publication, you may indicate that here to bypass the “Comments to the Author” section, enter your conflict of interest statement in the “Confidential to Editor” section, and submit your "Accept" recommendation.

Reviewer #3: (No Response)

2. Is the manuscript technically sound, and do the data support the conclusions?

Reviewer #3: Yes

3. Has the statistical analysis been performed appropriately and rigorously? 

Reviewer #3: Yes

4. Have the authors made all data underlying the findings in their manuscript fully available?

Reviewer #3: Yes

5. Is the manuscript presented in an intelligible fashion and written in standard English?

Reviewer #3: Yes

6. Review Comments to the Author

Reviewer #3: Interesting paper with only some issues, mainly related to abstract.

Abstract.

methods>proportion meta-analysis is not clear

methods>4 outcomes as primary outcome are too many. Authors should decide one primary outcome and other as co end points.

methods>time periods are not defined anywhere.

7. PLOS authors have the option to publish the peer review history of their article (what does this mean?). If published, this will include your full peer review and any attached files.

Reviewer #3: **Yes: **Fabrizio D'Ascenzo

---

## [Author Response · Author response to Decision Letter 1]

16 Oct 2020

Answers to the Reviewer #3

“Interesting paper with only some issues, mainly related to abstract. 

Abstract:

methods>proportion meta-analysis is not clear

methods>4 outcomes as primary outcome are too many. Authors should decide one primary outcome and other as co end points.

methods>time periods are not defined anywhere.”

Answer: We would like to thank you for taking the time to review our manuscript and the compliments on our study. We appreciated the suggestions and we incorporated them into the abstract, which we believe is now improved.

---

## [Decision Letter · Decision Letter 2]

21 Oct 2020

Perioperative and anesthesia-related cardiac arrest and mortality rates in Brazil: a systematic review and proportion meta-analysis

PONE-D-Leandro G. Braz 20-08860R2

Dear Dr.

We’re pleased to inform you that your manuscript has been judged scientifically suitable for publication and will be formally accepted for publication once it meets all outstanding technical requirements.

Kind regards,

Ehab Farag, MD FRCA FASA

Academic Editor

PLOS ONE

Additional Editor Comments (optional):

Reviewers' comments:

Reviewer's Responses to Questions

**Comments to the Author**

1. If the authors have adequately addressed your comments raised in a previous round of review and you feel that this manuscript is now acceptable for publication, you may indicate that here to bypass the “Comments to the Author” section, enter your conflict of interest statement in the “Confidential to Editor” section, and submit your "Accept" recommendation.

Reviewer #3: All comments have been addressed

2. Is the manuscript technically sound, and do the data support the conclusions?

Reviewer #3: (No Response)

3. Has the statistical analysis been performed appropriately and rigorously? 

Reviewer #3: (No Response)

4. Have the authors made all data underlying the findings in their manuscript fully available?

Reviewer #3: (No Response)

5. Is the manuscript presented in an intelligible fashion and written in standard English?

Reviewer #3: (No Response)

6. Review Comments to the Author

Reviewer #3: (No Response)

7. PLOS authors have the option to publish the peer review history of their article (what does this mean?). If published, this will include your full peer review and any attached files.

Reviewer #3: **Yes: **Fabrizio D'Ascenzo

---

## [Editor Report · Acceptance letter]

23 Oct 2020

PONE-D-20-08860R2 

Perioperative and anesthesia-related cardiac arrest and mortality rates in Brazil: a systematic review and proportion meta-analysis 

Dear Dr. Braz:

I'm pleased to inform you that your manuscript has been deemed suitable for publication in PLOS ONE. Congratulations! Your manuscript is now with our production department. 

Kind regards, 

on behalf of

Dr. Ehab Farag 

Academic Editor

PLOS ONE